# Recent Advances in the Control of Clinically Important Biofilms

**DOI:** 10.3390/ijms23179526

**Published:** 2022-08-23

**Authors:** Katarzyna Krukiewicz, Alicja Kazek-Kęsik, Monika Brzychczy-Włoch, Marek J. Łos, Collins Njie Ateba, Parvaneh Mehrbod, Saeid Ghavami, Divine Yufetar Shyntum

**Affiliations:** 1Department of Physical Chemistry and Technology of Polymers, Silesian University of Technology, M. Strzody 9, 44-100 Gliwice, Poland; 2Centre for Organic and Nanohybrid Electronics, Silesian University of Technology, Konarskiego 22B, 44-100 Gliwice, Poland; 3Department of Inorganic Chemistry, Analytical Chemistry and Electrochemistry, Silesian University of Technology, 44-100 Gliwice, Poland; 4Biotechnology Centre, Silesian University of Technology, Krzywoustego 8 Street, 44-100 Gliwice, Poland; 5Department of Molecular Medical Microbiology, Chair of Microbiology, Faculty of Medicine, Jagiellonian University Medical College, Czysta 18 Street, 31-121 Krakow, Poland; 6Department of Pathology, Pomeranian Medical University, 71-344 Szczecin, Poland; 7Food Security and Safety Niche Area, North West University, Private Bag X2046, Mahikeng 2735, South Africa; 8Influenza and Respiratory Viruses Department, Pasteur Institute of Iran, Tehran 1316943551, Iran; 9Faculty of Medicine in Zabrze, University of Technology in Katowice, Academia of Silesia, 41-800 Zabrze, Poland; 10Research Institute of Oncology and Hematology, Cancer Care Manitoba-University of Manitoba, Winnipeg, MB R3E 3P5, Canada; 11Biology of Breathing Theme, Children Hospital Research Institute of Manitoba, University of Manitoba, Winnipeg, MB R3E 3P5, Canada; 12Department of Human Anatomy and Cell Science, University of Manitoba College of Medicine, Winnipeg, MB R3E 3P5, Canada

**Keywords:** anti-biofilm agent, bacterial biofilm, bacterial infection, medical device, antimicrobial resistance, antibacterial therapy, antibacterial coating

## Abstract

Biofilms are complex structures formed by bacteria, fungi, or even viruses on biotic and abiotic surfaces, and they can be found in almost any part of the human body. The prevalence of biofilm-associated diseases has increased in recent years, mainly because of the frequent use of indwelling medical devices that create opportunities for clinically important bacteria and fungi to form biofilms either on the device or on the neighboring tissues. As a result of their resistance to antibiotics and host immunity factors, biofilms have been associated with the development or persistence of several clinically important diseases. The inability to completely eradicate biofilms drastically increases the burden of disease on both the patient and the healthcare system. Therefore, it is crucial to develop innovative ways to tackle the growth and development of biofilms. This review focuses on dental- and implant-associated biofilm infections, their prevalence in humans, and potential therapeutic intervention strategies, including the recent advances in pharmacology and biomedical engineering. It lists current strategies used to control the formation of clinically important biofilms, including novel antibiotics and their carriers, antiseptics and disinfectants, small molecule anti-biofilm agents, surface treatment strategies, and nanostructure functionalization, as well as multifunctional coatings particularly suitable for providing antibacterial effects to the surface of implants, to treat either dental- or implant-related bacterial infections.

## 1. Introduction

Microorganisms such as bacteria, viruses, and fungi, reside in diverse environments where they must adapt to various stresses, including pH, temperature, ultraviolet light, nutrient depletion, and oxygen variations, among others [1,2]. To survive and colonize a niche, these microbes must first bind to a substratum where they arrange into complex communities called biofilms. Clinically important biofilms can be formed when a single or multiple bacterial species attach to a substratum and produce an extracellular matrix comprising exopolysaccharide (EPS), proteins, and extracellular DNA (eDNA), as illustrated in Figure 1 [3]. This process involves highly-coordinated cell-to-cell communication mediated by quorum sensing (QS), small RNA fragments, or cyclic diguanosine-5′-monophosphate (c-di-GMP) and resulting in a coordinated gene expression associated with the growth of the biofilm [4,5,6]. The bacteria within biofilms are then (partially) protected from environmental factors, such as altered pH, osmolarity, extreme temperature, high pressure, or scarcity of nutrients, but also mechanical and shear forces [7]. Additionally, the biofilm structure protects bacteria from ultraviolet radiation, disinfectants, antibiotics, as well as from the host’s immunity factors [8].

The clinical problem associated with microbial biofilms is that they can cause persistent infections which are difficult to treat with antimicrobials. In other words, biofilms are recalcitrant or tolerant to different antibiotics, which is not the same as resistance. Because these terminologies have been used interchangeably in some review papers, it is therefore important that the authors defined the context in which they have been used in this review. Antibiotic resistance is most often related to mutations or the exchange of antibiotic-resistance genetic elements (acquired resistance), although resistance may also be intrinsic and thus dependent on wild-type genes and the innate properties of the cell [9,10]. Tolerance is the ability of a microorganism to survive, but neither grow nor die, in the presence of a bactericidal antimicrobial agent. Resistance can therefore be associated with both planktonic and biofilm bacteria, however, in this review, the authors will use the term tolerance/recalcitrance to describe hard-to-treat and persistent biofilms. In this light, biofilm tolerance can be associated with low susceptibilities to antibiotics due to slow growth rates, reduced oxygen and nutrient gradients, drug efflux pumps, conjugation, extracellular matrix-neutralizing proteins, and most importantly the presence of persister cells which are metabolically inert cells [11,12]. The low permeability of biofilms to antibiotics has been associated with EPS, which either binds and chelates the antimicrobial compounds, or degrades them using secreted enzymes [13]. Similarly, the limited diffusion of nutrients and oxygen within biofilms results in a heterogeneous population with fast-growing bacteria localized at the surface of the biofilm, and slow-growing, metabolically-inactive persisters in the interior of the biofilm [14,15]. Persister cells do not carry any genetic mutations and will revert to antibiotic-susceptible levels similar to the wild-type parental strain.

Small colony variants (SCV) are another type of metabolically inert bacteria found in some bacteria biofilms. SCV arise as a result of genetic mutations, and like persisters, represent a transient antibiotic-tolerant phenotype and revert to a killing curve resembling that of the wild-type parent upon re-exposure to the antibiotics [16]. Because several antibiotics, such as β-lactam antibiotics, target actively growing bacteria, slow-growing bacteria will tolerate these antibiotics, which enables their persistence and recolonization [17,18]. Furthermore, several studies have shown that phenotypic variants, i.e., persistent and small-colony variants (representing a sub-population of metabolically inert bacteria found in biofilms), can survive lethal concentrations of antibiotics without any known resistance mechanism or genetic change [19,20].

This article reviews the scientific literature on selected biofilm-associated diseases, namely dental- and implant-related infections (Figure 2). The overarching aim of this review is to highlight dental- and implant -associated biofilm infections that fall within the broad spectrum of implant-associated biofilm infections

However, because biofilms associated with implants can also result in secondary infections, the review will highlight advances in other pharmacological treatment options with a broader application to other biofilm infections such as the use of antibiotics, proteolytic enzyme, antimicrobial peptides, inhibitors of bis-(3′-5′)-cyclic dimeric guanosine monophosphate, and quorum sensing, hence showing the applicability of the recent advances in the pharmacology of antimicrobials.

## 2. Biofilm Infections

### 2.1. Dental Infections

Dental infections can be caused by several factors (e.g., changes in diet, oral hygiene, or antimicrobial use) which alter the homeostasis in the host’s mouth microbiome. These changes lead to the proliferation of pathogenic bacteria that may be associated with diseases, such as dental caries and periodontitis [21]. Dental caries (i.e., cavities or decay) are characterized by the demineralization of the teeth and are often associated with the fermentation of simple sugars, such as sucrose and lactose, by streptococci and lactobacilli [22]. The fermentation of these sugars produces acid, which erodes the enamel as well as the underlying dentin or connective tissue. If left untreated, caries may develop into inflammatory infections, e.g., pulpitis or apical periodontitis [23]. In pulpitis, the infection is localized in the root canal of the tooth and does not pass into the bone, whereas apical periodontitis causes inflammation and the destruction of periradicular tissues (e.g., root cementum, periodontal ligament, and alveolar bone) eventually leading to total pulp necrosis [24]. Dental caries has been associated with different bacterial genera, families, genus, and species based on culture-based and culture-independent 16 s metagenomic or short-gun sequencing. Some of the bacteria identified from the aforementioned studies include *Streptococcus mutans*, Actinomyces, Lactobacillus, Dialister, Eubacterium, Olsenella, Bifidobacterium, Atopobium, Propionibacterium, Scardovia, Abiotrophia, Selenomonas, and Veillonella, including carbohydrate-fermenting oral streptococci [25].

Periodontal diseases, e.g., gingivitis and periodontitis, are chronic inflammatory diseases involving tissue around the teeth [26]. Inflammation of the gingiva (gum) is characterized by redness, swelling, and bleeding from the crevice following mechanical stimulation. If left unchecked, gingivitis can lead to periodontitis, which causes more serious and irreversible damage to the teeth and surrounding tissue. In short, untreated gingivitis induces the development of pockets between the gums and teeth (periodontal pockets) leading to irreversible destruction of the periodontal ligament, alveolar bone, and cementum, which ultimately leads to tooth loss [27]. The subgingival microbiome is composed of *Actinomyces* spp., the mitis-group of streptococci, *Fusobacterium nucleatum*, *Veillonella parvula*, *Capnocytophaga* spp., *Rothia aeria*, *Rothia dentocariosa*, *Corynebacterium matruchotii*, and *Corynebacterium durum* [28]. The main periodontal pathogens include *Aggregatibacter actinomycetemcomitans*, *Porphyromonas gingivalis*, *Tannerella forsythis*, *Treponema denticola*, *F. nucleatum*, *Prevotella intermedia*, *Parvimonas micra*, *and Eubacterium nodatum* [29]. It is estimated that dental infections impact over 3.5 billion people globally, with most cases occurring in economically underdeveloped countries [30]. In addition to caries and periodontitis, several systemic infections have been associated with bacteria found in oral biofilms, including cardiovascular diseases, atherosclerosis, infective endocarditis, aspiration pneumonia, diabetes mellitus, preterm birth, and low-birth-weight babies [31,32]. Dental plaque or biofilms in the oral cavity contain dextran and glucan, which are products of dental pathogens, such as *S. mutans* and *Streptococcus sobrinus* [33]. Another cariogenic factor in dental caries is the production of glucan from sucrose by streptococcus. Glucans provide binding sites for pathogens to adhere to the surfaces of teeth.

### 2.2. Implant-Associated Infections

It is estimated that over 500,000 types of medical implants are available in the global market [34]. Some of these implants include cerebrospinal shunts, urinary and vascular catheters, cardiovascular electronic devices, breast implants, tracheal cannulas and tubes, contact lenses, dental fillings, prosthetic joints, and artificial ligaments [34]. It is common for exogenous or host-associated bacteria to form hard-to-eradicate biofilms on the implants, resulting in implant infections. In addition, invasive implants can trigger a localized inflammatory reaction, neutrophil activation, and a concomitant release of reactive oxygen species (ROS), which drastically reduce the ability of the host’s immune system to ward off pathogens [35]. Implant-associated bacterial infections are caused by a wide range of bacteria and depend on the implantation site and type of device. This review highlights two examples of implant-associated infections, namely, catheter and joint-related infections, in order to highlight the pathogenesis mechanisms as they relate to other medically important implants.

Catheter-derived urinary tract infections (UTIs) are primarily caused by *Proteus mirabilis*, *Proteus vulgaris*, and *Providencia rettgeri*. These bacteria produce urease, which hydrolyzes urea to ammonia and carbon dioxide [36]. Ammonia raises the urinary pH, thus inducing the precipitation of urinary salts as crystals. These crystals become trapped within the developing biofilms, where their growth is stabilized and enhanced by the biofilm matrix, ultimately generating a crystalline biofilm structure that blocks catheters [36]. If the blockage goes unnoticed, it can lead to the reflux of infected urine into the upper urinary tract and the onset of serious clinical complications, including pyelonephritis, septicemia, and shock [37,38]. In addition, the crystalline nature of these biofilms further contributes to the recalcitrance of these communities toward antibiotics [39]. The crystals can serve as a focal point for reinfection or for the formation of kidney or bladder stones, which can lead to further urinary obstruction [36]. Furthermore, uropathogens such as *Escherichia coli*, *Staphylococcus saprophyticus*, *Enterococci*, *Streptococci*, *Klebsiella pneumoniae*, *Pseudomonas aeruginosa*, and *Enterobacter* spp., release adhesion factors that are involved in catheter-associated biofilm development [38]. These biofilms may increase the ability of uropathogens to induce acute prostatitis and their persistence in the prostatic secretory system leads to recurrent UTIs characteristic of chronic bacterial prostatitis.

Orthopedic implant-related infections have been linked to joint arthroplasty (joint replacement) and osteosynthesis (bone repair) failure [40]. The main pathogens causing these infections include *Staphylococcus aureus*, coagulase-negative staphylococci (e.g., *Staphylococcus epidermidis*), and, to a lesser extent, *Cutibacterium*
*acnes* [34]. These bacteria use distinct mechanisms to attach to the implants, form biofilms, persist, and avoid a host’s defenses. In addition, the resulting biofilms are not only localized on the prosthetic, but can also spread to the synovial fluid, fibrous tissue, bone cement, and the bone itself, thereby increasing the bacteria’s ability to persist in the host and reinfect the implant [41]. This phenomenon is exemplified by the ability of *S. aureus* and *Staphylococcus lugdunensis* to invade osteoblasts, where they later recolonize the implants following lysis of the osteoblast [42]. Previous studies have also shown that the ability of *S. aureus* to enter the canaliculi of a live cortical bone contributes significantly to the recalcitrance of osteomyelitis [43]. Furthermore, the secretion of staphylococcal superantigen-like proteins 3 and 4 allows *S. aureus* to circumvent recognition by the host’s toll-like receptor 2 [44]. Staphylococcal biofilm infections can induce an anti-inflammatory response, characterized by the recruitment of anti-inflammatory macrophages and myeloid-derived suppressor cells, which suppress T-cell activation and prevent biofilm-associated phagocytosis [45].

## 3. Control Strategies for the Formation of Clinically Important Biofilms

### 3.1. Novel Antibiotics and Their Carriers

Conventional antibiotics administered either alone or in combination with other antibiotics have been shown to ameliorate the problem of antibiotic resistance (Figure 3). For example, it was reported that sub-minimum inhibitory concentrations (MICs) of ceftazidime reduce biofilm volume, inhibit twitching motility, and repress gene expression involved in bacterial adhesion and matrix production of *P. aeruginosa* [46]. Similarly, *E. coli* biofilms and planktonic cells were significantly reduced by colistin in a concentration-dependent manner [47]. An in vitro study [48] showed that 12 μg/mL of gentamycin released from bone graft substitutes could prevent *E. coli* adhesion, and 23 μg/mL of the antibiotic eliminated 24 h-old biofilms.

However, following the emergence of resistance, most antibiotics are administered in clinical settings in combination with other antibiotics. For example, although vancomycin remains the most commonly prescribed drug for *S. aureus* biofilm-associated infections [49], the emergence of vancomycin-resistant *S. aureus* has necessitated the provision of vancomycin together with other antibiotics, e.g., rifampin. Moreover, a combination of colistin with other antibiotics, such as tigecycline, has exhibited synergistic effects in vitro, thereby indicating their potential applicability in clinical settings [50,51]. It was demonstrated that amikacin, ciprofloxacin, and third-generation cephalosporins significantly reduced the biofilm biomass of several strains of *E. coli* associated with UTIs [52]. In addition, it was demonstrated that a combined treatment involving clarithromycin and daptomycin was useful to eradicate staphylococcal biofilms formed on titanium devices within 72 h [53]. The antibiotic lock technique (ALT) represents an adjunct therapy that can be used to treat catheter-related infections. The efficacy of an antibiotic lock solution (comprising meropenem, levofloxacin, and colistin) on clinical and reference *P. aeruginosa* strains was also confirmed [54]. It is important to note that although the use of combination therapy in the control of several infections is not novel in medicine, the rapid increase in antimicrobial resistance which has been observed over the past 20 years has necessitated continuous research into novel and effective drug combinations to fight both the antibiotic resistance of planktonic cells and recalcitrant biofilms to some drug combinations. This approach is therefore a mainstay in the fight against all clinical important biofilms.

#### 3.1.1. Antibiotic Adjuvants

Triclosan is a broad-spectrum antimicrobial agent that prevents type-II fatty acid synthesis in several bacterial species [55]. This compound has long been approved by the United States Food and Drug Administration (FDA) as a broad antimicrobial and antifungal agent used in toothpaste and other disinfectants. The combination of triclosan and tobramycin led to a 100-fold reduction in viable *P. aeruginosa*-persistent cells during 8 h of incubation, and resulted in complete eradication after 24 h; in contrast, triclosan alone had no appreciable effect [56]. The same study also showed that triclosan enhanced tobramycin’s efficacy in terms of killing multiple *Burkholderia cenocepacia* and *S. aureus* clinical isolates grown as biofilms. Additionally, triclosan exhibited synergy with gentamicin and streptomycin [56]. These findings demonstrate the potential application of adjuvants in the eradication of both dental and implant biofilms. However, with the exception of triclosan most antimicrobial compounds have not been approved for incorporation into hygiene products such as toothpastes, nor is there sufficient data on their bioavailability and efficacy in vivo.

#### 3.1.2. Antimicrobial Peptides and Proteins

Antimicrobial peptides (AMPs) are short (12–100 amino acids), cationic, and amphipathic molecules that are partially responsible for the innate immunity of bacteria, animals, and plants. Although most studies on AMPs have focused on their antibacterial properties, several reports have demonstrated their ability to inhibit biofilms, thus providing a potential therapeutic alternative [57]. The anti-biofilm activity of AMPs has been associated with their inhibition of attachment, the killing of planktonic cells, and/or eradication of mature biofilms [58].

Nisin is an FDA-approved and GRAS (generally recognized as safe) peptide with the recognized potential for clinical use. This antimicrobial peptide is a potent bactericidal agent against *Streptococcus pneumoniae*, *Clostridioides difficile*, and is even active against biofilms of methicillin-resistant *S. aureus* (MRSA) and biofilm formation involving several bacterial species [59]. In addition, nisin Z inhibits the growth of gram-negative oral pathogens, such as *P. gingivalis*, *P. intermedia*, *Aggregatibacter actinomycetemcomitans*, and *T. denticola* [60]. The authors also reported that nisin exerted anti-biofilm activity on saliva-derived multispecies biofilms without exhibiting cytotoxicity to human oral cells. A recent study [61] identified garvicin KS as a broad-spectrum AMP produced by *Lactobacillus garvieae*. This AMP showed bactericidal activity against 240 strains (19 species) of gram-positive bacteria, albeit limited activity against gram-negative bacteria [61]. Synthetic AMPs (e.g., Bac8c, HB43, P18, Omiganan, WMR, Ranalexin, and Polyphemusin) could successfully destroy *S. aureus* biofilms in a catheter [62]. However, there is no data showing that these synthetic AMPs will be successful in vitro. This not notwithstanding, AMPs represent an important alternative to antibiotics in the control of several biofilm infections including implant-associated infections. Furthermore, several in vitro studies have revealed antimicrobial and anti-biofilm properties associated with the synthetic peptides GH12, Lys-a1, and L-K6 against oral streptococci, such as *S. mutans*, *S. sobrinus*, and *Streptococcus salivarius* [63,64,65], thus highlighting their potential applications in dental care. Dental implants coated with antibacterial agents have been shown to limit bacterial growth, improve implant performances, and increase the success of dental treatment [66].

#### 3.1.3. Proteolytic Enzymes

Bacterial biofilms are usually attached to biotic or abiotic surfaces and are composed of polysaccharides, proteins, and nucleic acids. Therefore, compounds (protease/enzyme) that interfere with any one of these components will inhibit, or at least disrupt the EPS matrix, often leading to the detachment of the biofilm. For example, proteases (e.g., bromelain, actinidin, papain, proteinase K, and trypsin) have been reported to inhibit dental biofilms, including single-species and multi-species biofilms [67]. The anti-biofilm effects of DNAses have been demonstrated for organisms, such as *S. aureus*, *P. aeruginosa*, *E. coli*, *Acinetobacter baumannii*, *Haemophilus influenzae*, and *K. pneumoniae* [68]. Ficin, a sulfhydryl protease isolated from the latex of fig trees, was shown to disrupt *S. aureus* and *S. epidermidis* biofilms and enhance the anti-biofilm effects of antibiotics [69]. These findings show that proteases could be used to control biofilm infections associated with tooth and implant-associated infections.

The presence of dextran and glucan in dental biofilms has therefore inspired research into the use of glucanohydrolases, such as mutanase (α-1,3-glucanase) and dextranase (α-1,6-glucanase), as potential remedies or drugs for dental caries prevention [33,70]. A recent study demonstrated that dextranase or mutanase alone displayed limited efficacy in degrading biofilms; however, a synergistic effect was observed when both enzymes were used simultaneously [71]. A chimeric glucanase comprising mutanase and dextranase was shown to prevent dental biofilm formation by *S. sobrinus* [72]. Together, these findings support the possibility of introducing dextranases and glucanases in oral care products such as toothpaste and mouth wash.

#### 3.1.4. Bacteriophage Therapy

Phage-based therapy is a therapeutic alternative or adjuvant to antibiotics in the control of clinically important biofilms. Phage-based treatments include single phages or phage cocktails, phage-derived enzymes, phages in combination with antibiotics, and genetically modified ones [73]. For instance, Waters et al. [74] demonstrated that phage PELP20 caused a 3-log reduction in *P. aeruginosa* CFU in phage-treated biofilm (after 24 h). In addition, phage-encoded lysins, such as lysin CF-301 and LysH5, have been shown to be potent Staphylococcal antibiofilm agents based on their ability to degrade bacterial peptidoglycan [75,76]. Also in vitro and in vivo zebrafish infection models demonstrated that the chimeric lysin Csl2, obtained by fusion of the catalytic domain of Cp1-7 lysozyme to the CW-7 repeats of the LySMP lysine from a *Staphylococcus suis* phage, was able to remove *S. suis* biofilms [77]. Because biofilms are usually multispecies, phage cocktails containing two to five different phages have been used to increase antibiofilm efficiency. For example, a phage cocktail formulated by Maszewska et al. [78] to treat catheter-associated urinary tract infections caused by *Proteus mirabilis*, was able to destroy preformed biofilms and prevented biofilm formation. Phages have also been used in combination with antibiotics to eradicate biofilms. For example, Tkhilaishvili et al. [79] showed that *S.*
*aureus*-specific bacteriophage Sb-1 could eradicate biofilm, both alone and synergistically with different classes of antibiotics, degrade the extracellular matrix, and target persister cells of MRSA. 

### 3.2. Antiseptics and Disinfectants

A popular antiseptic and disinfectant, chlorhexidine is an effective antimicrobial compound that inhibits the growth of bacteria, fungi, and viruses [80]. This drug has therefore been incorporated into dental products such as oral rinses, aerosols, dental flosses, gels, and dental varnishes, and it is effective against caries, gingivitis, and plaque formation [80]. In addition, a synergism between chlorhexidine (2%), trypsin, and proteinase K was observed for reducing bacterial viable counts and disrupting nutrient-stressed biofilms (*C. acnes*, *S. epidermidis*, *Actinomyces radicidentis*, *Streptococcus mitis*, and *Enterococcus faecalis*) grown in the root canals of single-rooted teeth [81]. The incorporation of the above-mentioned cocktail into dental products could help control dental infections associated with biofilms. However, for this to be possible, the effectiveness of such a formulation will have to be demonstrated in clinical trials and the shelf life and bioavailability of these proteases also need to be experimentally determined.

Another type of disinfectant is quaternary ammonium salts (QAS). The presence of a hydrophilic ammonium cation and a hydrophobic alkyl chain enable QAS to interact with the cellular membranes of various microorganisms, including bacteria, fungi, viruses, and tumor cells to cause their death [82]. Unfortunately, since QAS are commonly used as cleaning agents, many microorganisms have developed resistance to these compounds. An interesting alternative to conventional QAS is so-called gemini surfactants (GS), which are composed of two monomeric QAS molecules connected by a spacer between the hydrophilic heads or the hydrophobic tails. Such GS have recently been studied as powerful bactericidal and fungicidal agents [82,83,84] because they exhibit very low critical micelle concentration, surface tension, and minimal inhibitory concentration. Labena et al. [83] synthesized a cationic GS (4,4′-(((1E,5E)-pentane-1,5diylidene) bis(azanylylidene))bis(1-dodecylpyridin-1-ium)bromide) and confirmed its significant antimicrobial, antibacterial, anti-candida, antifungal, anti-biofilm (anti-adhesive), and bio-dispersion properties. It is worth noting that the minimum inhibitory concentrations were as low as 0.004 mM for *S. aureus*, 0.04 mM for *E. coli*, and 0.15 mM for *Candida albicans*, respectively. Apart from their bactericidal and fungicidal activity, GS have demonstrated significant corrosion inhibition efficiency, particularly against the environmental sulfidogenic bacteria cultivated in a corrosive high-salinity medium [83].

### 3.3. Small Molecule Anti-Biofilm Agents

Although antibiotics have been employed to treat bacterial infections for centuries, their excessive use commonly causes alterations in intestinal and oral microbial flora and contributes to the development of bacterial resistance. Additionally, the efficiency of an antibiotic treatment is generally limited by the compact structure of a biofilm, which hinders drug penetration. Recent research results have suggested the use of small molecules to control the formation of a biofilm at the molecular level [85]. In general, the molecular mechanisms involving these compounds include either inhibition of the initial steps of biofilm growth, interfering with c-di-GMP signaling, or inhibition of quorum sensing (Figure 4).

#### 3.3.1. Inhibition of Biofilm Formation by Pilicides and Curlicides

Bacterial attachment is an initial step in the biofilm formation process and an essential step in most bacterial infections. To allow epithelial surface attachment, bacteria have evolved complex pili and fimbriae systems [86]. The structure of pili is highly conserved, and the subunits include adhesin, tip fibrillium, adaptor subunits, pilus base, termination, and anchoring units [87]. Apart from pili, some bacteria, e.g., *E. coli* and other Enterobacteriaceae, produce curli, which are adhesive amyloid fibers present at the surface of bacterial cells that are known to be critical for biofilm development [88]. Therefore, both pili and curli represent perfect targets to modulate the adhesion of bacteria and the development of biofilms.

FimH is the mannose-binding adhesin found at the tip of the type 1 pilus produced by most of the Enterobacteriaciae family, including uropathogenic *E. coli* (UPEC). This protein is an adhesin that has been associated with bacterial adhesion to biotic and abiotic surfaces, responsible for biofilm formation, proliferation, invasion, and internalization into eukaryotic cells and biofilm formation [89,90]. These properties have made FimH a prime target in the development of anti-adhesive therapeutic strategies which limit attachment and proliferation of bacteria expressing it. To this end, several α-d-mannose derivatives have been developed and used to produce glycomimetic drugs with enhanced affinity for FinH, improved stability, and bioavailability [91,92,93,94]. For example, using in vivo mouse models of UTI, Cusumano et al. [94] demonstrated that the aryl mannoside they developed (with in vitro antibiofilm properties) tightly binds FimH and prevents acute UTI, treats chronic UTI, and potentiates the efficacy of existing antibiotic treatments against *E. coli* strains, which are particularly important for the bacterial infections associated with the implantation of urinary catheters. In addition, monoterpene phenol carvacrol and its isomer, thymol, have been shown to exhibit strong curlicide or pilicide activity against uropathogens. Their activities were revealed when 79 essential oils were screened for antibiotic activity against uropathogenic *E. coli*. The results demonstrated high anti-biofilm and anti-virulence activities in carvacrol-rich oregano oil and thymol-rich thyme red oil [95]. Both carvacrol and thymol significantly inhibited biofilm formation at sub-inhibitory concentrations (<0.01%). The antibiotic activity of these compounds was explained by their ability to reduce fimbriae production and the swarming motility of bacteria, likely by disrupting curli and/or pili. Together, these findings suggest that the incorporation of curlicides and pilicides either into dental hygiene products or on the surface of implants may represent a potential therapeutic strategy in the control of several biofilm infections, however, this needs to be experimentally verified.

#### 3.3.2. Interfering with c-di-GMP Signaling

Bis-(3′-5′)-cyclic dimeric guanosine monophosphate (c-di-GMP) is a bacterial second messenger molecule that governs the motility, virulence, and cell cycle of some gram-negative and gram-positive bacteria [96,97]. It is known that high levels of c-di-GMP enhance bacterial adhesion by reducing the expression or activity of flagella and stimulating the production of bacterial adhesives and EPS. Therefore, controlling c-di-GMP metabolism could be an efficient way to modulate the formation of biofilm structures.

In recent work, a family of bacteriophage-encoded peptides that can be used to modulate c-di-GMP signaling in *P. aeruginosa* was identified and characterized [98]. Their mechanism of action involves direct interaction with diguanylate cyclase YfiN, an inner membrane protein identified as a key contributor to intracellular c-di-GMP levels in various bacteria [99]. The resulting increase in c-di-GMP production decreased the bacterial motility and increased the biofilm mass. Demonstrating that interference with intracellular signaling could regulate bacterial metabolism suggested that modulating the c-di-GMP signaling pathway could also be a good approach to interfering with biofilm formation. These expectations were confirmed by the development of a short peptide exhibiting nanomolar affinity and high specificity to c-di-GMP (19 residues). By directly interfering with a central signaling pathway, the developed peptide effectively inhibited the formation of biofilms in *P. aeruginosa.* This strategy is particularly interesting for the control of most biofilm infections in different parts of the body and requires more research. It also has the advantage over AMPs in that it does not kill the bacteria. This is particularly important in the case of dental biofilms because although a given compound might be effective in the control of dental biofilms it might also have other off-target effects ultimately resulting in dysbiosis in the mouth and the gut with clinical implications.

#### 3.3.3. Inhibition of Quorum Sensing

Bacterial QS is a cell-to-cell communication channel in which small molecules (particularly acyl-homoserine lactones, autoinducing peptides, and autoinducer-2) are used to coordinate bacterial behavior to help microorganisms survive [100]. By controlling bacterial virulence, QS plays an important role in the formation of biofilms. Therefore, chemical agents that can disturb bacterial signaling (anti-QS agents) are expected to reduce bacterial virulence without imparting drug resistance to the pathogens. Among potential alternatives to antibiotics, recent studies have indicated that organic acids (particularly acetic acid, citric acid, and lactic acid) show activity against *E. coli* and *Salmonella* sp. biofilms isolated from fresh fruits and vegetables [101]. Lactic acid achieves the maximum inhibition of violacein production among the investigated compounds, thus highlighting its efficiency against QS. As little as 2% lactic acid was able to reduce the production of violacein by 37.7%, leading to substantial reductions in biofilm formation, EPS production, and motility of bacteria. An additional advantage of organic acids is the fact that they are approved as GRAS by the FDA.

Another agent that combines the benefits of no adverse effects to humans, low price, and wide availability with an ability to disrupt the QS mechanism is ascorbic acid. Its efficacy in treating multidrug-resistant *E. coli* biofilms isolated from meat samples was recently reported [102]. Ascorbic acid (at a minimum inhibitory concentration of 125 mM) caused a 1.5-fold reduction in QS activity, a 5.8-fold reduction in EPS production, and a downregulation in the gene expression of the *luxS* and *bssR* genes, which are responsible for the production of QS effector molecules. Ascorbic acid’s ability to mediate the generation of reactive oxygen species meant that QS was further diminished following the alkaline hydrolysis of effector molecules.

Additionally, synthetic lactones have been shown to influence the QS mechanism of bacteria, particularly *P. aeruginosa*. It was hypothesized that it is possible to disable a QS system and prevent biofilm formation using synthetic lactones (e.g., *N*-(4-{4-fluoroanilno}butanoyl)-l-homoserine lactone and *N*-(4-{4-chlororoanilno}butanoyl)-l-homoserine lactone) that resemble homoserine lactone (HSL), a compound used in bacterial signaling [103]. Data from molecular docking experiments revealed that both synthetic lactones were able to bind to the active site of the ligand-binding domain of HSL receptors, thereby reducing biofilm formation by disabling the QS system without affecting the bacterial growth. As earlier mentioned for c-di-GMP inhibition, this antibiofilm strategy has the potential to control several different biofilm infections and its usefulness and application are limited by a lack of in vivo studies and outputs from clinical trials within the context of biofilm control. Therefore, more research needs to be conducted in this area to be able to ascertain the full clinical potential of this biofilm strategy.

### 3.4. Surface Treatment and Nanostructure Functionalization

Grinding, polishing, sand-blasting, sintering, heat treatment, and etching methods are often applied to modify the roughness and wettability of the surfaces of implants (Figure 5) [104,105]. A surface pretreatment is also performed before applying functional coatings to a metal surface. The physicochemical properties of the treated metal surface can be manipulated to alter bacterial adhesion and biofilm formation. It was reported that electrolytic polishing of the Ti-6Al-4V titanium alloy (which is widely used in medicine) is favorable to obtain a surface roughness of 10.33 nm ± 1.14 [106]. After the electropolishing process, the water contact angle was 92°, and the surface strongly inhibited the adhesion of *S. mutans*. When the surface roughness increased to 120.05 nm ± 7.89, the number of bacteria increased almost three times.

Metal surfaces can also be etched; for example, titanium is etched in a solution of hydrochloric acid and sulfuric acid at 98 °C [107], giving an average surface roughness of the etched Ti of 2.67 μm ± 0.20. The same Ti alloy was anodized at 40 V in a solution containing glycol ethylene and NH_4_F at 300 °C, and the surface roughness decreased to 1.07 μm ± 0.09. Differences in *S. epidermidis* adhesion between these two samples were detectable, but not significant; in this case, the nanostructures on the surface affected the number of adhered bacteria. The effect of surface topography on *S. epidermidis* adhesion on a Ti surface was also evaluated [108], and it was found that polished and plasma-sprayed surfaces were less covered with bacteria than grit-blasted or satin-treated surfaces. The surface chemical composition also influences bacterial interactions with the surface. Therefore, Ti surface treatments are being investigated to determine optimal surface roughness and chemical composition for osteoblast-like cell adhesion and antibacterial properties. So far, the best results were obtained for a plasma-sprayed Ti surface. Similar work was presented in 2020 [109]; however, they analyzed a Ti-Cu alloy and confirmed that, in addition to the surface morphology, the chemical composition of the material plays a key role in *S. aureus* bacteria colonization and activity.

Apart from providing antibacterial effects, physical treatment often enhances the biocompatibility of surfaces. The increase in roughness, for instance, enhances the interactions between the implant surface and the tissue by offering additional adhesion sites for cells [110]. This is particularly important for dental and bone implants that need to exhibit osseointegration (the tight connection between bone cells and implant surface) to resist repetitive mechanical loading.

### 3.5. Surface Modification Strategies

Biofilm development on the surfaces of medical implants is a severe problem that has increased the frequency of required revision surgeries, and sometimes, fatalities. Therefore, recent investigations have focused on selecting and testing various surface modification strategies to prevent bacterial colonization on medical surfaces. Because the antibacterial activity of surfaces depends mainly on the chemical composition and the morphology of the surface, recent research efforts have been particularly devoted to engineering surface roughness and developing miscellaneous biologically-active coatings (Figure 6) [111].

#### 3.5.1. Engineering Surface Roughness and Topography via Laser Treatment

The ability of bacteria to attach to a surface is mainly governed by the physical interactions between the surface and the cell wall of a microorganism. Numerous studies aimed at modulating bacterial attachment by tailoring surface roughness, wettability, and topography [112], have shown that bacterial attachment can generally be reduced by introducing surface features smaller than microorganisms [113]. It was shown that, in some cases, fine surface features reduced bacterial retention more effectively than extremely smooth surfaces. Recent studies have also demonstrated the suitability of laser surface treatment technologies for introducing nano-textures to the surface of medical implants, particularly orthopedic, thus providing anti-adhesive effects. For example, the surfaces of commercially pure Ti (Grade 2) and Ti_6_Al_4_V (Grade 5) alloy implant materials were successfully modified [114]. A laser surface treatment performed using a fiber laser with a near-infrared wavelength (1064 nm) was applied to provide rosette-like markings on the metallic surface, consisting of secondary micro-/nano-sized features, such as ripples and radial lamellae. As a result of this surface engineering, a significant increase in the arithmetic mean roughness (up to 27 times) and a maximum height of the profile (up to 20 times) were observed. Analysis of the bacterial adhesion using *S. aureus* as a model bacterial strain demonstrated a noticeable bactericidal effect on the laser-treated metallic surfaces, which was mainly attributed to the presence of nano-features, a reduction in the surface hydrophobicity, and the formation of stable oxide films [114].

The effect of laser nitriding on the biological activity of a beta titanium alloy was investigated in 2020 [115]. A 1064-nm laser was also used in this study, but instead of rosette-like features, the surface was engineered with crescent structures having dendritic patterns emanating from the center of the crescent shapes. Interestingly, the arithmetic mean roughness and the maximum height of the profile only experienced five-fold increases in this case. Nevertheless, the surfaces subjected to laser nitriding exhibited lower tendencies to be covered by a bacterial (*S. aureus*) biofilm than the analogous unmodified surface, especially in the early stages of attachment. This antibacterial character was associated with the appearance of unique surface patterns, an increase in roughness, and the hydrophilic nature of laser-nitrided surfaces.

Lutey et al. [116] made an important contribution to the design of laser-textured antibacterial surfaces by fabricating spikes, laser-induced periodic surface structures (LIPSS), and nano-pillars on the surfaces of stainless steel specimens, analyzing their antibacterial properties against *E. coli* and *S. aureus*. The samples differed in terms of the peak separation between surface structures, their average areal surface roughness, and their wettability. The results showed increased adhesion of *E. coli* to the surfaces characterized by dimensions exceeding the cell size (e.g., mirror-polished specimens and spikes). However, the same surfaces were found to inhibit the adhesion of *S. aureus*, mainly because of their low surface roughness (characteristic of mirror-polished samples) and low wettability (characteristic of spikes). Both LIPSS and nano-pillars reduced the retention of both bacterial strains but were more effective toward *E. coli* than *S. aureus*. The reduced adhesion rate was attributed to the low wettability and fine surface features (similar in size to bacterial cells) associated with LIPSS and nano-pillars, which were expected to limit the number of available attachment points. Since the use of ultrashort pulsed laser processing enables the facile tailoring of both wettability and surface morphology, it is expected that further advances in this technology will make it possible to fabricate the next generation of antibacterial surfaces without affecting their biocompatibility with mammalian cells.

#### 3.5.2. Biomimetic Anti-Adhesion Coatings

Research on surfaces that prevent bacterial adhesion has increased tremendously within the last couple of years. Current interest in this field is focused on the development of anti-adhesion coatings inspired by natural surfaces, e.g., cicada wings, gecko skin, shark skin, dragonfly wings, and plant leaves [117], or derived from natural sources [118]. For example, biomimetic hierarchical diamond films were developed to mimic the morphology of plant leaves and exhibited self-cleaning, antibacterial, and anti-biofouling properties, together with mechanical and chemical stability [119]. They applied a bottom-up strategy based on a hot-filament chemical vapor deposition to fabricate micro- and nano-sized diamond coatings on the surface of Ti_6_Al_4_V, silicon, and quartz glass. In each case, diamond films reduced bacterial attachment (using *E. coli* as a model strain) by 90–99%. Due to the biocompatibility of diamond and the mechanical robustness of diamond-based coatings, it is expected that the developed materials should be applicable for the modification of orthopedic implants. Mussels inspired the development of multifunctional dendritic polyglycerol (MI-dPG)-based coatings [120]. To mimic the adhesive mussel foot proteins, MI-dPG was functionalized with amines and catechol moieties. By controlling the surface polymerization conditions, it was possible to obtain hierarchical micro- and nano-meter rough surfaces exhibiting super-hydrophobic, super-hydrophilic, and even super-amphiphobic wetting properties. Interestingly, the super-hydrophilic (water contact angle close to 0°) and super-amphiphobic (water contact angle of 173° ± 1°) surfaces prevented bacterial attachment most effectively, i.e., by 99.7% and 96.3%, respectively. To eliminate *E. coli* and *S. aureus* “survivors”, MI-dPG coatings were loaded with silver nanoparticles, which led to remarkable antibacterial behavior (>99.99% antibacterial activity) and potential applicability as implant coatings.

#### 3.5.3. Multifunctional Antibacterial Coatings

Recent advances in biomedical engineering opened the possibility of developing novel materials that exhibit multifunctional characteristics. For decades, scientists have been interested in designing surfaces with bacteriostatic or bactericidal properties, and numerous strategies have been applied for this purpose, including antibacterial agents (antibiotics, biocides, metals, enzymes, organic cationic/non-cationic compounds, etc.), contact-killing coatings, or bacteria-repelling surfaces [121]. Having achieved great success in the design of bacteriostatic and bactericidal coatings, recent research efforts have shifted toward designing highly-functional materials that, in addition to their activity towards bacteria, possess other advantages suited to their final applications. Therefore, novel antibacterial coatings are engineered to possess multifunctional characteristics, usually including the ability to support the growth and differentiation of eukaryotic cells.

Orthopedic implants are highly susceptible to bacterial infections. The greatest challenge in the development of multifunctional implant coatings is to find a balance between anti-adhesive properties (desired for interactions with bacteria) and pro-adhesive properties (necessary for proper osteointegration). Hoyos-Noguyes et al. [122] improved conventional Ti surfaces by introducing cell-adhesive and antibacterial properties. The resulting surfaces of the Ti implants were coated with an anti-fouling layer of poly (ethylene glycol) (providing a bacteriostatic effect) and then biofunctionalized through the simultaneous immobilization of two peptides: RGD (enhancing cell adhesion) and LF1-11 (exhibiting bactericidal activity). This trifunctional coating effectively inhibited (by 99.8%) the initial adhesion of bacteria (*Streptococcus sanguinis*) and promoted the adhesion of osteoblasts (human sarcoma osteogenic cells, SaOS-2), achieving a two-fold increase in the number of cells attached to the surface. By eliminating two major reasons for implant failure, i.e., bacterial infection and poor osteointegration, the developed approach has significant potential for orthopedic medicine and dentistry.

Tissue-implant interactions are often extremely complex and must therefore be analyzed with a broad perspective. For example, it is well established that the promotion of osteointegration is essential for the successful implantation of bone substituents. However, it is also important to consider angiogenesis, since this process plays a crucial role in maintaining homeostasis with the regenerated bone tissue. Accordingly, a tantalum copper composite nanotube coating was developed, which was found to improve the bacteriostatic, angiogenic, and osteogenic properties of titanium implants [123]. The bacteriostatic properties of the coating (tested against *E. coli* and *S. aureus*) were provided by copper ions that were released from the coating in a slow and stable manner, achieving a steady-state release rate after 21 days. It was also hypothesized that if the copper release is a slow, long-term process (as was noted in the presence of tantalum), then the released ions should be able to activate endothelial receptors and promote angiogenesis. Indeed, investigated materials enhanced cell migration, tube formation, and the expression of angiogenesis genes (fibroblast growth factor receptor 1, endothelial nitric oxide synthase, and vascular endothelial growth factor) in the in vitro model of human umbilical vein endothelial cells. Further tests performed with classic osteogenesis in vitro model (mouse embryo osteoblast precursor cells, MC3T3-E1) confirmed the osteogenic properties of tantalum copper composite nanotube coatings, as evidenced by the increased extracellular matrix mineralization. Additionally, the materials exhibited higher corrosion resistance than a titanium control surface by reducing the release of Ti^4+^ ions when exposed to bacteria or an acid environment.

In addition to their antibacterial activity and cytocompatibility, Pang et al. “armored” their coatings with another property, namely piezoelectricity [124]. Because living bone can be considered a piezoelectric tissue, piezoelectricity is expected to contribute to bone growth and bone remodeling, as well as cell behavior at the molecular level (i.e., via gene expression, protein synthesis, cell differentiation, and proliferation) [125]. ZnO was selected as the material that could embody the desired biological functions. ZnO is a typical piezoelectric material, and its nanoparticles have recently been shown to possess both antibacterial and bone growth activities [126]. ZnO/TiO_2_ coatings were fabricated by a hydrothermal synthetic technique followed by a low-temperature liquid phase method, resulting in structures that resembled a nanoarray of nanoparticles. These ZnO/TiO_2_ coatings achieved maximum bacteriostatic activities of 99% against *S. aureus* and 90% against *E. coli*, and they promoted the proliferation of osteoblast precursor cells (MC3T3-E1) and the expression of alkaline phosphatase (indicating the presence of osteoblast cells and the formation of new bone tissue). To analyze the piezoelectric properties of the coating on osteogenesis, cell cultures were subjected to a periodic loading (6 N traction tension applied at a frequency of 0.25 Hz for 4 s over a period of 1, 4, or 7 days) in a biomechanical reactor. The results revealed a 50% increase in the proliferation rate and a two-fold increase in the alkaline phosphatase activity when piezoelectric materials were used, compared with the non-piezoelectric control (both subjected to periodic loading). Another recent study [127] indicated that such antibacterial effects could be controlled by modulating the piezoelectric properties of the coatings.

#### 3.5.4. Electroactive Coatings

It is known that biofilm bacteria use electrical signaling for communication. Therefore, it is reasonable to expect that they are also susceptible to electrical stimulation. Indeed, recent studies [128] have revealed that electrochemical methods are powerful techniques to expand the scope of conventional strategies for biofilm modulation. Although the precise mechanism of bacterial growth inhibition has not yet been fully elucidated, it is suspected that an electrical current with an adequate current density might be able to either damage bacterial membranes or block the multiplication of bacterial cells. Another possible mechanism involves the indirect effect of an electrical current on bacterial cells, which would be mediated by electrically-induced changes in pH, temperature, or the generation of toxic products during electrolysis. Accordingly, designing electroactive coatings represents a promising approach for controlling biofilm growth. Conducting polymers, particularly, have recently gained a research interest as versatile biomaterials with numerous bioengineering applications, including regional chemotherapy [129,130], tissue scaffolds [131], neural interfaces [132,133], and controlled drug delivery systems for antibiotics [134].

Gomez-Carretero et al. [135] presented an interesting insight into understanding the interactions between bacteria (*Salmonella enterica* serovar Typhimurium) and electrochemically-active surfaces (poly(3,4-ethylenedioxythiophene); PEDOT). Specifically, they showed that it was possible to examine biofilm formation on the surface of PEDOT in the reduced and oxidized states through external polarization. They also determined that the growth of the biofilm was supported on the surface of oxidized PEDOT-coated electrodes and diminished (by 52–58%) on the surface of reduced PEDOT-coated electrodes. Studies regarding bacterial growth on “unswitched” PEDOT-coated electrodes have revealed similar reductions in biofilm formation as those observed on the surface of the reduced polymer, suggesting that the presence of bacteria was responsible for the electrochemical reduction in PEDOT. Therefore, conducting polymer surfaces can be treated as smart antibacterial implant coatings that exhibit antibacterial activity in the presence of bacterial cells.

The antibacterial effects of conducting polymers can be further enhanced through the immobilization of silver nanoparticles [136], as in the multilayer coating comprising a poly(ethyl terephthalate) electrode covered with poly(3,4-ethylenedioxythiophene):polystyrene sulfonate (PEDOT:PSS) and (3-aminopropyl) triethoxysilane. The latter allowed for the binding of silver nanoparticles by forming coordinate bonds with amine groups. The antibacterial efficacy of the composite coating, assessed against *S. aureus*, was significantly enhanced when the material was electrically stimulated (5 Hz, square wave voltage varying from −2 to 2 V), ultimately achieving a reduction in biofilm growth close to 90%. Interestingly, without electrical stimulation, the reduction in biofilm growth on the surface of the composite only reached 50% and no change in biofilm growth was observed for the polymer control (i.e., not functionalized with silver nanoparticles). It was clear that electrical stimulation enhanced the antibacterial effect of silver nanoparticles, thus verifying a new, effective approach to controlling biofilm colonization on medical implants.

Conducting polymers can also be employed as antibiotic carriers that enable electrically-triggered controlled drug release and electrically-driven antibacterial activity. In a recent study [137], a PEDOT matrix was used to immobilize a powerful first-line antibiotic (tetracycline; Tc). Although a portion of the immobilized Tc was released spontaneously, the electrical trigger (a chronoamperometric potential jump from –0.6 V to –0.5 V, applied for 2 s and 600 s, respectively) made it possible to control the overall rate of drug release. Interestingly, antibacterial activity against *E. coli* was observed for both PEDOT and PEDOT/Tc, which reduced the bacterial cell density by 40% and 55%, respectively. Additionally, bacterial cells grown on the surface of both polymers were evidently smaller than those grown on a control surface, suggesting a decrease in the metabolic activity of the bacteria due to the presence of PEDOT or PEDOT/Tc. Therefore, similar to a previous report [135], it was expected that the bacteria could partially reduce PEDOT, thereby inducing its antibacterial activity and allowing additional Tc to be released. Thus, a conducting polymer-based coating was again confirmed as a potential candidate for a self-adaptive antibacterial system suitable for surface modification of medical implants.

#### 3.5.5. Switchable Coatings

A “holy grail” in the pursuit of controlling biofilm formation would be developing a surface that could detect bacterial colonization and prevent it spontaneously, without human intervention. This approach can be realized using smart materials that respond (i.e., significantly change their characteristics) to an external stimulus, which could be oxygen stress, moisture, light, temperature, pH, etc. [138,139]. The crucial issue in the development of intelligent antibacterial coatings would be to select a stimulus associated with the presence of bacteria that would be strong enough to trigger the desired changes in the material.

Because acidic environments are always generated in bacterial infections, pH was used as a trigger to induce surface charge-switchable properties in a novel nanocomposite composed of silver nanoparticles decorated with carboxyl betaine groups exhibiting zwitterionic nature [140]. Under physiological conditions (pH 7.4), the carboxyl groups were deprotonated, and the material exhibited good cytocompatibility with healthy tissue. Decreasing the pH value to 5.0 or below resulted in the protonation of carboxyl groups, and the positively-charged surface of the silver nanoparticles enhanced their interactions with bacteria. It was also shown that protonated silver nanoparticles could penetrate into the structure of the biofilm and kill the bacteria (*E. coli* and *S. aureus*) deep within the biofilm. Therefore, by modifying the surfaces of silver nanoparticles with zwitterionic structures, it was possible to form a coating that exhibited good cytocompatibility when in contact with healthy cells but was able to switch to a strong antibacterial agent when the environmental conditions (acidity) changed as a result of a bacterial infection. However, because the bacteria-induced changes in pH can be moderate, e.g., the median pH associated with abdominal/anorectal infections is ~6.75 [141], the next challenge facing the design of pH-triggered switchable materials should involve enhancing their sensitivity to this environmental trigger.

Another approach to designing “smart” antibacterial surfaces is also based on silver nanoparticles, but in this case, they were embedded in temperature-responsive matrices comprising poly(di(ethylene glycol)methyl ether methacrylate) and poly(4-vinylpyridine) [138]. These two polymers are known to have wetting behaviors that can change significantly following a relatively small change in temperature; this effect is caused by the disruption of hydrogen bonds between the polymers and water when the temperature exceeds the low critical solution temperature (LCST) of the material. Accordingly, poly(di(ethylene glycol)methyl ether methacrylate) undergoes a sharp transition of its water contact angle (from 30° to 50°) at a temperature of around 19 °C. Similar behavior was also observed for poly(4-vinylpyridine), whose water contact angle changed from 30° to 45° at around 12 °C. In addition to the changes in the water contact angle, such changes in temperature altered the release behavior of silver ions. For example, at temperatures below the LCST, silver ions were blocked inside the polymer matrix, but temperatures above the LCST resulted in the collapse of the polymer structure and a concomitant release of silver ions with antibacterial activity. Additionally, the increase in temperature above the LCST was expected to enable the direct surface binding of silver nanoparticles to the bacteria, which was prevented below the LCST. Consequently, biological tests with *E. coli* and *S. aureus* as model bacterial strains revealed the temperature-switching ability of the fabricated materials, which triggered their antibacterial activity. At 37 °C there were almost no living bacterial cells present on the surface of both temperature-responsive coatings, which was not the case at 4 °C. Thus, polymer-based coatings with self-activating temperature-dependent properties solidified their potential as switchable antibacterial coatings for various biomedical applications, able to induce antibacterial effects upon implantation.

#### 3.5.6. Antibacterial/Anti-Adhesive Porous Oxide Layers

Plasma electrolytic oxidation (PEO) is widely used to form porous oxide layers on various metal surfaces, including titanium, aluminum, magnesium, niobium, tantalum, and their alloys [142,143,144,145]. These materials have been found to be applicable to the design of short-term or long-term bone implants. As a result of the plasma electrolytic process, a porous layer is formed on the surface of the metal, with pores in the range of several nanometers to several micrometers. The pore size strongly depends on the parameters applied during the anodization process, such as the voltage, current density, bath and chemical composition of the treated metals, and duration of the process. The surface pretreatment (e.g., grinding, polishing, sand-blasting, or etching) also influences the final porous oxide layer. Microstructures on the oxide layer are favorable for bone tissue formation; therefore, this process has potential applicability as a surface treatment for long-term implants.

Owing to the lack of antibacterial properties in a TiO_2_ porous layer, the coatings are enriched with antibacterial agents, or the agents are deposited on top of the layer through another technique. A one-step surface treatment is achieved when the oxide layer is formed on the electrode during the anodization process, and the antibacterial chemical compounds are incorporated from the anodizing bath. Porous oxide layers can be treated using additional techniques, e.g., electrophoretic deposition, ion implantation, atomic layer deposition, or the widely-used dip coating process [146]. Then, the resulting functional hybrid coating is equipped with antibacterial properties to protect the surface against the adhesion of bacteria, the formation of biofilms, and in some cases, biologically-active substances are released from the coatings to treat bone-related infections [147]. Porous oxide layers might also be immersed in a solution containing biologically-active substances, e.g., in a solution of betamethasone sodium phosphate, which exhibits antimicrobial properties [148]. Copper and zinc ions are recognized as antibacterial agents for such purposes, especially because they decrease the adhesion of bacteria and the formation of biofilms. Copper- and zinc-based compounds were successfully incorporated into an oxide layer on a Ti surface when the PEO process was carried out in anodizing bath containing Zn(CH_3_COO)_2_ and Cu(CH_3_COO)_2_ [149].

However, the addition of an antibacterial agent influences the cytocompatibility of the coatings. For example, the deposition of Ag on a porous oxide layer significantly decreased the number of MC3T3-E1 cells relative to the coating with a deposited Pt compound [150]. It was also reported that NaF may show antibacterial activity when added into an anodizing bath and incorporated into the oxide layer as an F-based compound [151]. The main goal of treating the metal implant surface is to introduce the functionalization according to their medical requirements. Therefore, it is necessary to find a balance between the antimicrobial activity of the metal surface and its cytocompatibility. Table 1 presents examples of antibacterial agents incorporated into the porous layer (formed via PEO) or deposited on top of the oxide layer using various techniques. The antimicrobial activity of these surfaces was usually evaluated against *S. aureus* or *E. coli*, rather than against clinically important bacteria strains. Table 1 also presents information about the results of antimicrobial analysis and cytocompatibility evaluations.

## 4. Conclusions

The formation of bacterial biofilms and the bacterial resistance to antibiotics have been recognized as critical challenges facing modern medicine. These problems are further complicated by the emergence of multidrug-resistant pathogens whose eradication now requires the use of last-resort antibiotics. Nevertheless, antibiotic therapy remains a viable treatment option for controlling chronic infections arising from biofilms. In addition to antimicrobial chemotherapy, other strategies include the use of compounds that degrade the matrix, inhibit cell-to-cell signaling, increase susceptibility of the biofilms to antimicrobial compounds and phagocytosis, or in the case of implants, remove/replace contaminated implants or change their physicochemical properties.

The surface treatment of metal implants is one method used to protect the implants against bacterial adhesion and biofilm formation. Current research efforts are focused on developing nanostructured surfaces with adhered/incorporated or melted anti-adhesive agents. Still, inorganic compounds based on copper, silver, or zinc compounds represent good potential solutions when designing functional surfaces with anti-adhesive bacterial properties. In particular, copper-based compounds exhibit excellent antibacterial properties and better cytocompatibility than silver-based compounds. Future investigations into surface treatments should involve the incorporation of antibiotics or other natural chemical compounds. However, the limited stabilities of these sensitive compounds indicate that inorganic compounds could find wider applicability in biomaterials. It is anticipated that the number of studies related to the production of novel materials destined for medicine should and will increase. The exploitation of the biocompatibility of ceramic, metal, and polymer materials or composites with antibacterial properties is one future research direction that will undoubtedly optimize biomaterials technology.

## Figures and Tables

**Figure 1 ijms-23-09526-f001:**
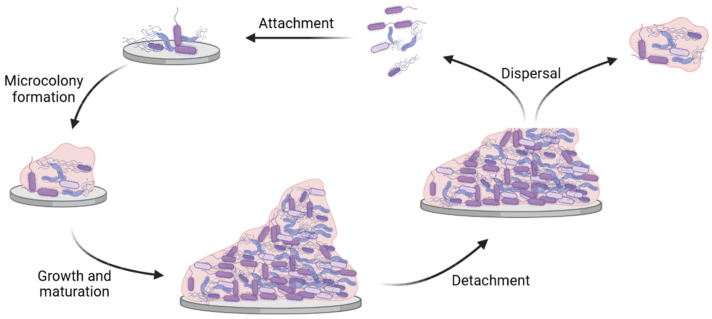
Schematic representation of the formation and dispersal of bacterial biofilm. Attachment represents the initial phase of biofilm growth, during which planktonic bacteria bind to an abiotic or biotic surface. Thereafter, microcolonies are formed when irreversibly adhered bacteria grow as a multicellular community. Proliferation and secretion of extracellular matrix components leads to the growth and maturation of the biofilm. Finally, detachment and dispersal of biofilms results in the release of both planktonic and microcolonies to the environment.

**Figure 2 ijms-23-09526-f002:**
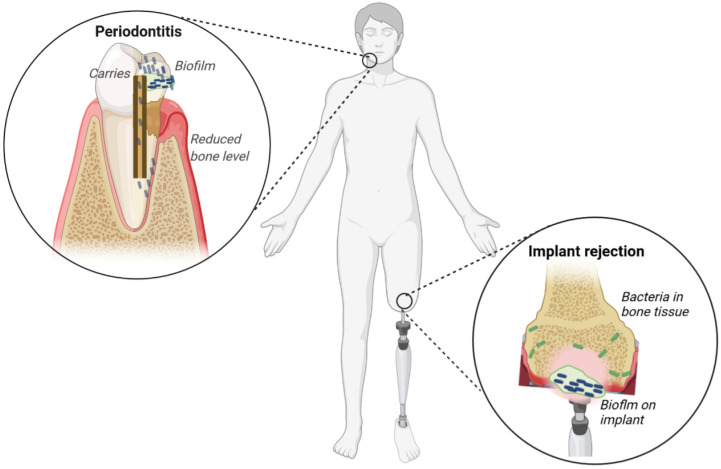
Schematic representation of some clinically important biofilms. Periodontitis and carrier are examples of dental infections resulting from a change in the homeostasis between the host and mouth microbiome, leading to the proliferation of pathogenic bacteria. Implant rejection is a highly undesired endpoint of a bacterial infection from either exogenous or host-associated bacteria.

**Figure 3 ijms-23-09526-f003:**
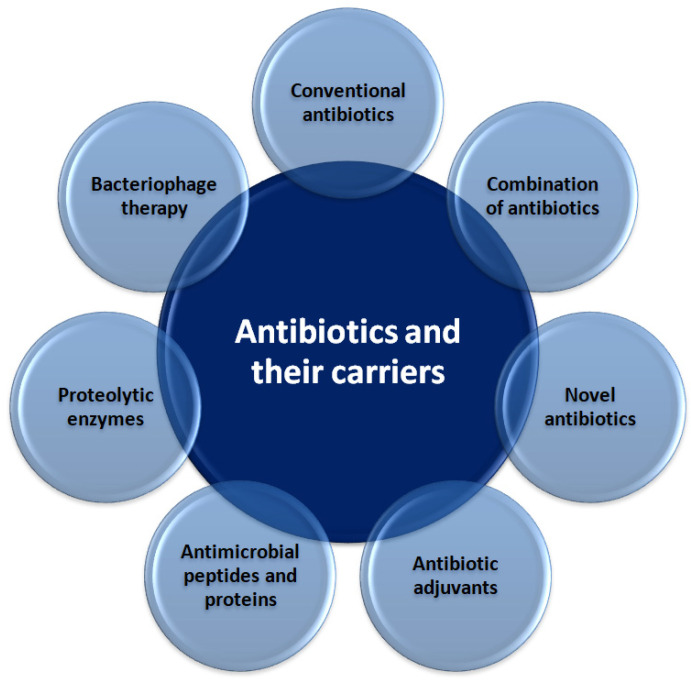
Schematic representation of antibiotic-based strategies for controlling the formation of clinically important biofilms.

**Figure 4 ijms-23-09526-f004:**
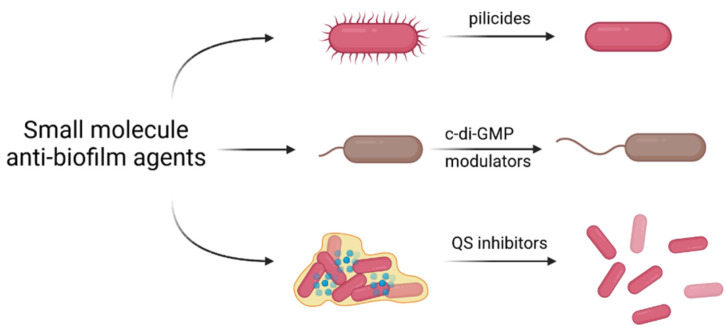
Schematic representation of anti-biofilm mechanisms of small molecules, including inhibition of biofilm formation by pilicides and curlicides, interfering with c-di-GMP (bis-(3′-5′)-cyclic dimeric guanosine monophosphate) signaling, or inhibition of quorum sensing (QS).

**Figure 5 ijms-23-09526-f005:**
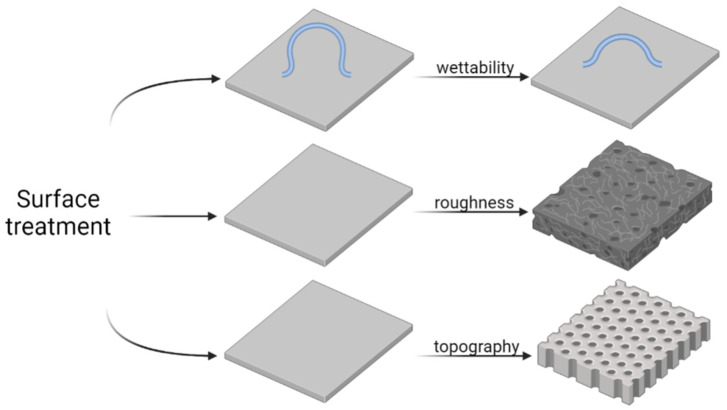
Schematic representation of surface treatment strategies to modulate bacteria adhesion and biofilm formation by changing surface properties, namely wettability, roughness, and topography.

**Figure 6 ijms-23-09526-f006:**
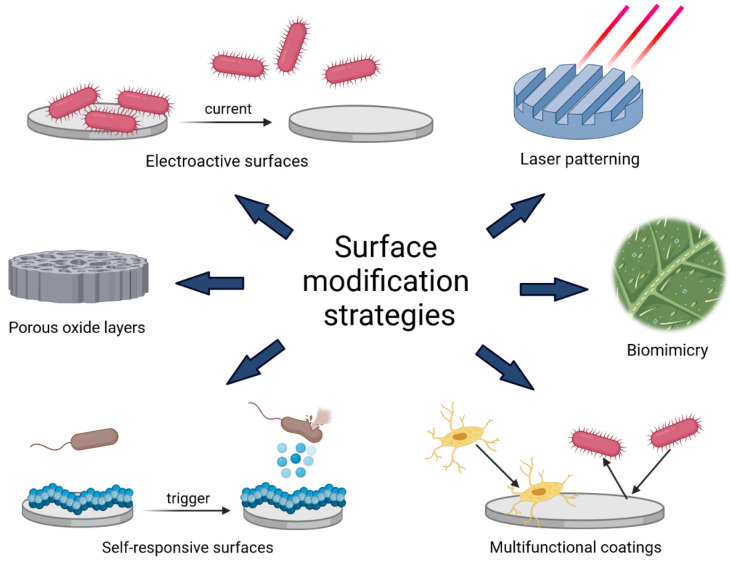
Schematic representation of surface modification strategies allowing the control of biofilm growth on the surface of implants, namely electroactive surfaces, surfaces engineered by laser treatment, biomimetic anti-adhesion coatings, multifunctional antibacterial coatings, switchable and self-responsive antibacterial coatings, and antibacterial/anti-adhesive porous oxide layers.

**Table 1 ijms-23-09526-t001:** Antimicrobial effects of various porous oxide layer coatings formed via plasma electrolytic oxidation on metal surfaces or the effect of hybrid coatings on microbial activity when the base layer is formed by anodization. The cytocompatibility of the coatings is evaluated to determine the balance between the cytotoxicity of various bacteria and the acceptable cytocompatibility of treated surfaces.

Antibacterial Agent	Type of Coating	Bacterial Strain	Cytocompatibility Analysis	Comments	Ref.
Cu_2_O, ZnO	TiO_2_ with incorporated Cu_2_O and ZnO	*E. coli* (CMCC (B) 44102)	n.a. *		[152]
CuO, Cu_3_(PO_4_)_2_	TiO_2_ with incorporated CuO, Cu_3_(PO4)_2_	*S. aureus* ATCC 25923 *E. coli* ATCC 25922	osteoblast-like MG-63 cells	anti-adhesive properties	[144]
Cu_2_O, CuO	Al_2_O_3_ with incorporated Cu_2_O, CuO	sulfate reducing bacteria	n.a.	anti-biofilm formation properties	[153]
Ag, Pt	hybrid coating TiO_2_-Ag or Pt deposited by ion implantation	*S. aureus* 839 and 224/228 (methicillin-resistant), *E. coli* U20 (antibiotic-sensitive) and K261 (antibiotic-resistant)	osteoblast MC3T3-E1 subklon 4 cell	anti-adhesive properties	[154]
Ag NPs, Zn NPs, Pt NPs	TiO_2_ with incorporated selected NPs or mixture of NPs	*S. aureus* MRSA USA300	MC3T3-E1 cells	-	[155]
Ag nanoparticles	TiO_2_ with incorporated Ag nanoparticles	*E. coli* ATCC 25922, *S. aureus* ATCC 6538	n.a.	-	[156]
Ag, Ag_2_O NPs	TiO_2_ with incorporated Ag and Ag_2_O NPs	*S. aureus* ATCC 6538	MC3T3-E1 cells	-	[157]
AgNO_3_	TiO_2_ with incorporated Ag compounds	*E. coli* ATCC 25822	n.a.	-	[158]
ZrO_2_, ZnO	deposition of Zr on Ti surface by pulsed direct current (DC) magnetron sputtering and then anodization	*S. aureus*, ATCC6538	MC3T3-E1 cell	-	[159]
ZrO_2_	TiO_2_ with incorporated ZrO_2_	*P. aeruginosa*, *E. coli*	n.a.	-	[160]
ZnO NPs	TiO_2_ with incorporated ZnO	*S. aureus* ATCC 25923, *E. coli* ATCC 25922	n.a.	-	[161]
Zn(CH_3_COO)_2_	TiO_2_ with Zn-based compound and hydrothermal treatment	*S. aureus* ATCC 25923, *E. coli* ATCC 25922	n.a.	-	[162]
Na_2_WO_4_	TiO_2_ with incorporated W-compounds	*E. coli*, *S. aureus*	n.a.	-	[163]
Na_2_WO_4_	TiO_2_ with incorporated W-compounds	*E. coli*, *S. aureus*	n.a.	-	[164]
Al_2_O_3_	anodized Al alloy in H_2_SO_4_ solution	*E. coli* ATCC 25922	n.a.	-	[165]
graphene oxide	hybrid coating: TiO_2_-graphen oxide deposited by EPD	*E. coli* DM 3423, *S. aureus* DM 346	n.a.	anti-adhesive properties	[166]
Ta_2_O_5_	hybrid coating TiO_2_-Ta_2_O_5_ deposited by high-power impulse magnetron sputtering	*S. aureus*, *Actinobacillus actinomycetemco-mitans*	human skin fibroblasts (HSF) and human osteosarcoma cells MG-63	-	[150]
C_6_H_9_O_6_Y	TiO_2_ with incorporated Y_2_O_3_	*S. aureus* ATCC 25923, *E. coli* ATCC 25922	fibroblast	-	[167]
ZrO_2_	MgO with incorporated ZrO_2_	*E. coli* PTCC 1330	n.a.	-	[168]
Na_2_B_4_O_7_	TiO_2_ with boron-based compounds	*S. aureus*, *P. aeruginosa*	Adipose derived stem cells (ADSC)	-	[169]

* n.a.—not analyzed; NPs—nanoparticles.

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
