# Peer review of "Recent Advances in the Control of Clinically Important Biofilms"

_ijms, 2022, doi:10.3390/ijms23179526_

Round 1

Reviewer 1 Report

In my opinion the manuscript "Recent advances in the control of clinically important biofilms" is an interesting and comprehensive piece of work. It is also well-written and clear. The schemes are of good quality and support the information given in the text. 

My suggestions:

1. Please avoid repeating "form/formed/formation" (e.g. lines 24 and 25; 43 and 44...).

2. Line 48: Please add "QS" after quorum sensing.

3. Lines 50-51: Biofilms are protected not only from the host immunity and antibiotics but also from other factors like disinfectants and UV radiation. Please complete the information.

4. I suggest separating the information given in the paragraph (lines 156-178) as an independent subsection 2.1.1.

5. In the Table 1 the information in the column "Antibacterial effect" is not consistent. Antibacterial properties, antibacterial activity, antibacterial ability, antibacterial effect mean more or less the same, hence, I don't see the point in using different expressions here and in my opinion it should by unified. Moreover, in my opinion, this column is not necessary, because it does not contain any additional information in many rows. I suggest:

1. Deleting "antibacterial properties, antibacterial activity, antibacterial ability, antibacterial effect" and substitute them with "not described" with "N.D." shortage in the table (or something similar). OR

2. Changing the column name into "Comments" (or similar), moving it after Cytocompatibility analysis column and giving only important information there, like: "prevents the formation of sulfate reducing biofilm". 

I hope that the authors understood my intention. 

Author Response

In my opinion the manuscript "Recent advances in the control of clinically important biofilms" is an interesting and comprehensive piece of work. It is also well-written and clear. The schemes are of good quality and support the information given in the text. 

My suggestions:

  1. Please avoid repeating "form/formed/formation" (e.g. lines 24 and 25; 43 and 44...).

As suggested by the Reviewer, we have rephrased mentioned lines. Additionally, we have checked the rest of the manuscript to ensure that there are no unneccessary repetitions.

  1. Line 48: Please add "QS" after quorum sensing.

As suggested by the Reviewer, we have added QS abbreviation after “quorum sensing” term in line 48.

  1. Lines 50-51: Biofilms are protected not only from the host immunity and antibiotics but also from other factors like disinfectants and UV radiation. Please complete the information.

According to the suggestion of the Reviewer, we have completed the required information.

  1. I suggest separating the information given in the paragraph (lines 156-178) as an independent subsection 2.1.1.

We would like to thank the Reviewer for this suggestion. Taking into account the comment of the second Reviewer, we have decided to move the content of the above-mentioned paragraph to the relevant parts of Section 3: Control strategies for the formation of clinically important biofilms.

  1. In the Table 1 the information in the column "Antibacterial effect" is not consistent. Antibacterial properties, antibacterial activity, antibacterial ability, antibacterial effect mean more or less the same, hence, I don't see the point in using different expressions here and in my opinion it should by unified. Moreover, in my opinion, this column is not necessary, because it does not contain any additional information in many rows. I suggest:
  2. Deleting "antibacterial properties, antibacterial activity, antibacterial ability, antibacterial effect" and substitute them with "not described" with "N.D." shortage in the table (or something similar). OR
  3. Changing the column name into "Comments" (or similar), moving it after Cytocompatibility analysis column and giving only important information there, like: "prevents the formation of sulfate reducing biofilm". 

I hope that the authors understood my intention. 

We would like to thank the Reviewer for this comment. Indeed, we agree that the information presented in the column labelled “Antibacterial effect” was not consistent. Therefore, we have changed it into the column labelled “Comments”, and provided only the necessary information.

Reviewer 2 Report

The review provides a broad and detailed description of methods for combating the formation and destruction of biofilms, but there are some remarks:

1. The abstract of the article does not correspond to the content of the article itself. It is not stated that this article reviews the scientific literature on selected biofilm-associated diseases, namely dental and implant-related infections. In addition, most of the review is devoted to control strategies for the formation of clinically important biofilms.

2. Introduction (p. 2 line 82) you are talking about the second group of metabolically inert bacteria, but the description above does not indicate which bacteria belong to the first group

3. Dental caries have been associated with bacteria such as Streptococcus mutans, Actinomyces, Lactobacillus, Dialister, Eubacterium, Olsenella, Bifidobacterium... (on page 3). You talk about bacteria, but you specify both bacteria and species of genera.

4. Why write about antimicrobial substances in the description of dental infections (p. 4), especially since they are mentioned in paragraph 3.1.3 (p. 7)?

5. F. nucleatum listed twice in one sentence (p. 4). The names of microorganisms in the text of the article are given in full somewhere, and somewhere in the abbreviation. It is necessary to write the full name of the microorganism if it is mentioned in the text for the first time, and if it is mentioned again, it can be abbreviated. For example Streptococcus mutans and S. mutans.

6. Is it possible to use the methods you described in the treatment and prevention of biofilm formation in related to dental and implant-related infections? What is their biocompatibility? If not, then perhaps they do not need to be indicated in this review.

Author Response

The review provides a broad and detailed description of methods for combating the formation and destruction of biofilms, but there are some remarks:

  1. The abstract of the article does not correspond to the content of the article itself. It is not stated that this article reviews the scientific literature on selected biofilm-associated diseases, namely dental and implant-related infections. In addition, most of the review is devoted to control strategies for the formation of clinically important biofilms.

We would like to thank the Reviewer for their comments. According to the suggestion of the Reviewer, we have changed the abstract to better match the body of the manuscript.

  1. Introduction (p. 2 line 82) you are talking about the second group of metabolically inert bacteria, but the description above does not indicate which bacteria belong to the first group

We would like to thank the Reviewer for this comment. Indeed, it was not clear that the first group of metabolically inert bacteria are persister cells as described in the previous paragraph. Therefore, we have rephrased the sentence to make it clear to the Readers.

  1. Dental caries have been associated with bacteria such as Streptococcus mutans, Actinomyces, Lactobacillus, Dialister, Eubacterium, Olsenella, Bifidobacterium... (on page 3). You talk about bacteria, but you specify both bacteria and species of genera.

We would like to thank the Reviewer for this comment. We have corrected the used terms.

  1. Why write about antimicrobial substances in the description of dental infections (p. 4), especially since they are mentioned in paragraph 3.1.3 (p. 7)?

We would like to thank the Reviewer for this comment. Not to mention antimicrobial substances suitable for the treatment of dental infections twice, we have decided to move the content of the above mentioned paragraph (page 4) to the relevant parts of Section 3: Control strategies for the formation of clinically important biofilms (pages 7-9).

  1. F. nucleatum listed twice in one sentence (p. 4). The names of microorganisms in the text of the article are given in full somewhere, and somewhere in the abbreviation. It is necessary to write the full name of the microorganism if it is mentioned in the text for the first time, and if it is mentioned again, it can be abbreviated. For example Streptococcus mutans and S. mutans.

We would like to thank the Reviewer for the thorough evaluation of the manuscript. We have removed additional “F. nucleatum” name from the mentioned sentence, and checked the rest of the manuscript to be consistent with the names of bacteria. As suggested by the Reviewer, the full name of the microorganism has been written when it was mentioned in the text for the first time. If it was mentioned again, its name has been abbreviated.

  1. Is it possible to use the methods you described in the treatment and prevention of biofilm formation in related to dental and implant-related infections? What is their biocompatibility? If not, then perhaps they do not need to be indicated in this review.

We would like to thank the Reviewer for their concern. In fact, all approaches to control of clinically important biofilms mentioned in this review paper are related to either dental or implant-related infections. All described materials/compounds are biocompatible and suitable for medical purposes. We have made sure that this information is placed in the manuscript (page 1, lines 38-39). Also, we have highlighted the biomedical applicability of described methods where necessary (lines: 557-562, 628-629, 641-643, 654, 812, 861-862).